# Natural Alkaloids in Cancer Therapy: Berberine, Sanguinarine and Chelerythrine against Colorectal and Gastric Cancer

**DOI:** 10.3390/ijms25158375

**Published:** 2024-07-31

**Authors:** Anna Duda-Madej, Szymon Viscardi, Wiktoria Szewczyk, Ewa Topola

**Affiliations:** 1Department of Microbiology, Faculty of Medicine, Wroclaw Medical University, Chałubińskiego 4, 50-368 Wrocław, Poland; 2Faculty of Medicine, Wroclaw Medical University, Ludwika Pasteura 1, 50-367 Wrocław, Poland; szymon.viscardi@student.umw.edu.pl (S.V.); wiktoria.szewczyk@student.umw.edu.pl (W.S.); ewa.topola@student.umw.edu.pl (E.T.)

**Keywords:** alkaloids, anticancer, antitumor, berberine, chelerythrine, colorectal cancer, cytostatic, gastric cancer, natural compounds, sanguinarine

## Abstract

The rising incidence of colorectal cancer (CRC) and gastric cancer (GC) worldwide, coupled with the limited effectiveness of current chemotherapeutic agents, has prioritized the search for new therapeutic options. Natural substances, which often exhibit cytostatic properties, hold significant promise in this area. This review evaluates the anticancer properties of three natural alkaloids—berberine, sanguinarine, and chelerythrine—against CRC and GC. In vivo and in vitro studies have demonstrated that these substances can reduce tumor volume and inhibit the epithelial–mesenchymal transition (EMT) of tumors. At the molecular level, these alkaloids disrupt key signaling pathways in cancer cells, including mTOR, MAPK, EGFR, PI3K/AKT, and NF-κB. Additionally, they exhibit immunomodulatory effects, leading to the induction of programmed cell death through both apoptosis and autophagy. Notably, these substances have shown synergistic effects when combined with classical cytostatic agents such as cyclophosphamide, 5-fluorouracil, cetuximab, and erlotinib. Furthermore, berberine has demonstrated the ability to restore sensitivity in individuals originally resistant to cisplatin GC. Given these findings, natural compounds emerge as a promising option in the chemotherapy of malignant gastrointestinal tumors, particularly in cases with limited treatment options. However, more research is necessary to fully understand their therapeutic potential.

## 1. Introduction

The human organism in a state of health is characterized by homeostasis, a balance of cells between death and their divisions. When this state is disrupted, errors in cellular DNA recording and reading occur, resulting in the abnormal and uncontrollable growth of one population of cells. The formation of a mass of tissue called a tumor is a result of uncontrolled division [1].

In this review, we focused on gastrointestinal submucosal tumors (GIST), and more specifically discussed gastric (stomach) cancer (GC) and colorectal cancer (CRC), which rank first (55.6%) and third (6%), respectively, in terms of the incidence of this type of cancer [2]. From a medical standpoint, due to the lack of diagnostic criteria and a specific code in the International Classification of Diseases (ICD) system, they are the most common mesenchymal tumors of the gastrointestinal tract [3,4,5]. Additionally, they rank third (CRC) and fifth (GC) among cancers worldwide according to data from the World Cancer Research Fund International (WCRFI) [6,7]. CRC is predicted to increase its incidence to 3.2 million per year by 2040, with 1.6 million deaths per year (compared to 2020, when there were 1.9 million cases, including 930.000 fatalities) [8], which could contribute to an even higher place in the cancer ranking [9].

Age has been shown to have a correlation with both types of cancer according to scientific data, but gender has not been linked (except for a slight predominance among men without statistical significance) [10,11]. The American Cancer Society (ACS) reports that the risk of developing CRC is at a similar level in both sexes: 1/23 in men vs. 1/25 in women [12]. However, survival rates in young women have been shown to be significantly higher compared to older women and men [13,14,15]. Estrogen is believed to be the cause of this, as it regulates the activity of Kv channels (KCNQ1:KCNE3). At the same time, it was shown that activation of protein-coupled estrogen receptor (GPER) leads to inhibition of proliferation in CRC cells, promoting their apoptosis [16]. Johnson et al.’s study confirms its beneficial contribution to CRC, indicating that estrogen replacement therapy reduces the risk of CRC by up to 17%. Moreover, its duration acts in proportion to its protective effect [17]. 

It is also interesting to note that the geographic location of the two cancers discussed in this review is their common feature [17,18]. Many of the countries with the highest recurrence rates overlap for both GC and CRC [18,19]. According to the WCRFI, the leading countries (with the highest AGS, age-standardized rates per 100.000 inhabitants) for GC are Japan, Korea, Iran, China, Russia, and Vietnam, with ASR/100.000 ranging from 27.6 to 13.4 [7]. On the other hand, for CRC, the leading countries are Japan, France, the United Kingdom, Italy, Russia, the United States of America, and China (ASR/100.000 range 36.6 to 20.1) [6]. 

The fundamental cause of the tumorigenesis process is gene mutations in the DNA of healthy cells [20]. Capturing the damaged cell and inhibiting its growth until it is repaired or activates a self-destruct mechanism is the crucial role played by the immune system [21]. The production of a signaling protein chain that prevents the division of damaged cells is accomplished by a number of genes called suppressor genes. In the case of GIST, these include KIT/PDGFRA mutations, affecting the gene encoding receptor expression tyrosine kinase (*KIT*) and the receptor for platelet-derived growth factor alpha (*PDGFRA*) [22,23,24,25,26]. Next, these mutations initiate uncontrolled activation of cell signaling pathways. For GIST, these are: (i) PI3K (phosphatidylinostol 3-kinase)/AKT (protein kinase)/mTOR (mammalian target of rapamycin); (ii) RAS (rat sarcoma, monomer GTP)/RAF (rapidly accelerated fibrosarcoma)/MAPK (mitogen-activated protein kinase), and (iii) JAK (Janus-activated kinase)/STAT3 (signal transducer and activator of transcription) [27].

TKI therapy, which involves the inhibition of tyrosine kinases, is the main approach to treating GIST, taking into account its molecular basis. The most commonly used for this purpose are imatinib, sunitinib, and regorafenib [28,29]. The primary and secondary resistance of GIST to this therapy is influenced by the tumor’s molecular characteristics, despite the promising results of this treatment. Primary resistance, disease progression occurring within the first 6 months of treatment, is primarily associated with substitutions in the *PDGRFA* gene or duplication of the *KIT* gene [30,31,32]. On the other hand, secondary resistance, observed after many months or even years of TKI therapy, is associated with secondary mutations in the *KIT/PDGFKA* genes, which change the ATP uptake site or activation loop of the receptor [33,34,35]. Direct interference with surrogate kinases can also be used to overcome the existing resistance, leading to selective activation and inhibition of certain ones. The combination of imatinib with crizotinib (an MET (receptor for hepatocyte growth factor/ALK) inhibitor) or cabozantinib (a KIT/MET inhibitor) has shown very promising results in the treatment of GISTs [36]. 

Innovative biologic drugs are particularly promising for the treatment of advanced forms of CRC and GC. The Federal Drug Agency (FDA) has so far approved a number of monoclonal antibodies targeting colorectal cancer: (1) anti-EGFR (epidermal growth factor receptor), i.e., cetuximab, panitumumab [37]; (2) anti-BRAF (v-Raf murine sarcoma viral oncogene homolog B1), i.e., encorafenib [38]; (3) anti-KIT/PDGFR, i.e., regorafeni [39]; (4) anti-PD1 (programmed cell death, i.e., nivolumab [40]; (5) anti-CTLA-4 (cytotoxic T-lymphocyte-associated protein-4), i.e., ipilimumab [41,42]. 

The effectiveness of even the latest chemotherapy options is reduced by internal tumor resistance, which unfortunately limits the options for effective cancer treatment. Scientists are increasingly turning to naturally derived compounds in their quest to solve this problem. These substances can provide anticancer effects by the following mechanisms: (i) antioxidant/antioxidant activity (neutralization of excess free radicals, e.g., isoflavones, flavonoids, catechins) [43,44,45]; (ii) DNA repair (reducing the proliferation of abnormal cells, such as sulfur compounds of garlic) [46]; (iii) mutation protection (e.g., flavonoids, carotenoids, saponins, terpenoids, anthraquinones) [47]; (iv) inhibition or reduction of tumor growth (e.g., sulfur compounds of garlic) [46,48]; and (v) maintaining normal levels of apoptosis (e.g., isothiocyanates) [49,50]. Conscious and widely understood cancer prevention is enhanced by the knowledge of the benefits provided to us by nature for many thousands of years.

In this review, we focus on presenting the anticancer properties of three plant-derived alkaloids—berberine (BRB), chelerythrine (CHL), and sanguinarine (SAN) (Figure 1)—in relation to colorectal and gastric cancer therapy. All these substances are characterized by numerous biological, anti-inflammatory, anti-infective, and anticancer effects [51]. 

Berberine is an isoquinoline alkaloid commonly found in plants of the *Berberidaceae* family (leaves, stems, twigs, barks, rhizomes, and roots), e.g., *Berberis vulgaris*, as well as the genera *Mahonia* and *Coptis chinensis* (Franch) [52]. It has been shown in a number of studies that the substance is active against numerous cancers: HCC [53], NSCLC [54], ovarian cancer [55], breast cancer [56], prostate cancer [57], and glioblastoma multiforme [58], among others. Chelerythrine is also a plant-derived compound (benzophenanthridine alkaloid) occurring, among others, in *Toddalia asiatica* (Linn.) and *Chelodonium majus* L. It also has antimicrobial activity against drug-resistant bacterial strains, e.g., MRSA or ESβL. Antifungal activity has also been reported against, e.g., *Ustilaginoidea virens* and *Cochliobolus miyabeanus*. However, in this article, we focused on the anticancer effects of the mentioned alkaloid [59,60]. Sanguinarine is also a benzophenanthridine alkaloid of natural origin, primarily isolated from the roots of *Sanguinaria canadensis*, as well as from other plants belonging to the families *Papaveraceae, Fumariaceae*, and *Rutaceae* [61,62]. It has been utilized in traditional medicine due to its well-known antibacterial, antioxidant, and anti-inflammatory properties [63], while the anticancer activity of SG has attracted particular interest in the past years, becoming the subject of numerous studies involving multiple cell cancer types such as gastric, colorectal, osteosarcoma [64], breast [65], lung [66], melanoma [67], and others [61,62].

In writing this review, we focused in particular on the aspect of the molecular basis of the anticancer activity of the above-mentioned substances. Targeted oncological therapy that yields the best results can be developed by detecting molecular interference points in cytostatic preparations. However, this requires understanding how the substance affects cancer cells at the level of gene expression and signal cascade activity.

## 2. Activity of Natural Compounds against Colorectal Cancer

### 2.1. Berberine

In their study, Liu et al. showed significant properties of BRB in the field of inhibition of the development and metastasis of colorectal cancer (CRC). BRB has been shown in vitro to suppress the growth of cancer cells of the SW620 (25.83% inhibition) and LoVo (30.66%) lines (IC50 54.41 µM and 78.66 µM, respectively). Moreover, there was a significant decrease in the ability of CRC cells to migrate with relatively low systemic toxicity (in vivo murine model). The attenuation of cyclooxygenase 2 (COX-2) and prostaglandin E2 (PGE2) expression, which have been recognized for their roles in the development of CRC, was discovered by molecular analysis. BRB has been demonstrated to restrict the migration and invasiveness of CRC cells by silencing specific signaling pathways (JAK-2/STAT-3). The experiment also showed that overexpression of COX-2 and PGE2 can significantly activate the JAK-2/STAT-3 pathways, and thus, the ability of BRB to block the entire cascade COX-2/PGE2/JAK-2/STAT3 makes it an extremely promising option in CRC therapy [68]. A multicenter, double-blind, randomized clinical trial evaluated the ability of BRB to prevent the recurrence of colorectal adenoma (adenomas usually promotes the development of CRC in the future). In the study group, patients who had a history of polypectomy received 0.3 g of BRB orally twice daily. It has been shown that the use of BRB in CRC chemoprophylaxis is associated with a reduced risk of adenoma recurrence (36% of relapses in the BRB group and 47% in the placebo group) [69].

Yu et al. investigated how *Fusobacterium nucleatum* colonizing the large intestine during the early stage of CRC development affects the progression of the mentioned cancer. It has been proven (using a mouse in vivo model) that *F. nucleatum* significantly increases the percentage of cancer among tested animals and also adversely affects the intestinal microbiota. The study also assessed the effect of BRB on the *F. nucleatum*-dependent oncogenesis process. Restoring the balance of the colonic microbiota and reducing CRC development were associated with the use of BRB. Molecular analysis showed a decrease in the formation of pro-inflammatory cytokines (originally secreted in response to the colonization of *F. nucleatum*), including IL-21, IL-22, IL-31, and CD40L. In addition, it has been shown that BRB can inhibit the activity of JAK/STAT and MAPK/ERK signaling pathways in the CRC cell population by decreasing the secretion of the cytokines mentioned. Researchers have postulated that BRB, through its complex mechanism of action, can therefore inhibit CRC tumorigenesis mediated by *F. nucleatum* [70]. Li et al. investigated the anti-oncogenic properties of BRB in an in vivo murine CRC model. Significant decrease in BRB-treated cancer cells activity was demonstrated (60%, *p* = 0.009). The expression of COX-2 and Ki-67 decreased in CRC cells that were exposed to BRB. In the molecular analysis, the possible detection of BRB antitumor pathway based on AMPK-dependent inhibition of mTOR kinase and AMPK-independent inhibition of the NF-κB signaling pathway was performed. BRB exposure had a negative impact on the number of substrates for mTOR kinase, which were 4E-binding protein 1 and p70 ribosomal S6 kinase. There was also a decrease in the expression of survivin and cyclin D1, both of which are factors associated with development of colorectal cancer. The phosphorylation of p53 protein and the formation of cleaved caspase-3 in cancer cells were both induced by BRB [55].

Wu et al. studied the effectiveness of Gegen Qinlian decoction (which includes BRB as its main component, 491.4 mg/g) combined with irinotecan (a camptothecin derivative) in the treatment of chemotherapy-induced intestinal inflammation in an in vivo murine model. The authors reported that the decoction led to a significant decrease in the concentration of pro-inflammatory cytokines, including IL-1β and TNF-α. It was also revealed that COX-2 activity was significantly decreased. It is noteworthy that BRB and irinotecan have shown potential synergy in the field of tumor reduction, specifically in the HT-29 CRC cell assay. The preparation in combination with the cytostatic drug seems to be a very interesting option; at the same time, it leads to the intensification of the action of the camptothecin derivative while reducing the toxicity of therapy with its participation [71]. The influence of BRB and evodiamine as factors that may inhibit the process of CRC development by interaction with micro-RNA molecules miR-429 was also evaluated. Untreated cancer cells were found to have a high expression of miR-429 molecules with a simultaneous decrease in Par3 and E-cadherine expression, as demonstrated by the study. Loss of functional genes encoding E-cadherin and Par3 led to the disintegration of epithelial cells and loss of cell adhesion, which increased the invasiveness of the tumor. Therapy with BRB and evodiamine, in turn, contributed to the reduction of miR-429 expression, and the tumor’s ability to invade the stroma decreased as a result [72]. The Chen et al. study investigated the effect of BRB on the development of CRC and on the colonic microbiota in mice in which cancer development was induced in vivo. BRB reduced colon tumor growth, and on a molecular level, the decrease in Ki-67 expression was also confirmed. Using microbiological analysis, it was shown that BRB restored the proper appearance of intestinal microbiota, which are disturbed in the course of CRC (decrease in the percentage of *Bifidobacterium, Barnesiella, Odoribacter*, and opportunistic pathogens with an increase in the content of *Alloprevotella, Oscillibacter,* and *Parabacteroides*). The restoration of the microbiota resulted in the limitation of pro-inflammatory lipid metabolism in the intestinal mucosa area, which includes arachidonic acid, which is typical for tumorigenesis. A complex metabolomic analysis showed that BRB contributed to an increase in the production of SCFA by bacteria and reduced glycometabolism in the intestinal mucosa area, contributing to a decrease in CRC activity [73]. 

Similar reports come from the Chen et al. study. The anti-inflammatory and anticancer effects of BRB were associated with blocking the NF-κB signal cascade. Use of BRB in mice (in vivo model) was associated with significant remodeling of the colorectal microbiota, as observed in the above-mentioned studies [74]. Similar research has been conducted by Yan et al. to investigate the impact of BRB on the formation of SCFA and the remodeling of colonic microbiota. Researchers have also reported the existence of another potentially significant BRB–CRC cell molecular interaction: inhibition of the pro-inflammatory TLR4/NF-κB/p65/IL-6/STAT3 signaling pathway with simultaneous intensification of expression of proteins conditioning the connection of epithelial cells in the tissue area: occludin and ZO1 protein [75]. Another study evaluated how CRC cells respond to the combination of BRB and luminespib (NVP-AUY922) therapy. The Hsp90 inhibitor has a high activity against CRC cells characterized by hyperexpression of the aforementioned chaperone; however, it may induce secondary overexpression of the cytostatic-limiting protein called survivin, leading to the development of cancer resistance to chemotherapy. BRB’s ability to decrease survivin expression in cancer cells was demonstrated in the study; it has a synergistic effect with the Hsp90 inhibitor (in relation to HCT-15 and HT-29 cell lines). In relation to the HCT-15 line, it was shown that a combination of preparations resulted in a significant decrease in phosphorylation of the mTOR, p70S6K, AKT, and GSK3β pathways. The effect of BRB–cytostatic drug connection was seen in blocking CDK4 expression and suppressing the Pin1-β-catenin-cyclin D1 signaling pathway via micro-RNA-296-5p. Compared to cytostatic alone, the combination resulted in a more intense induction of miR molecule synthesis [76].

Li et al. evaluated the possible effect of BRB on the process of CRC development conditioned by colorectal inflammation (*colitis*). The study showed a significant decrease in the formation of proinflammatory cytokines in the area of the colon mucosa: IL-6 and TNF-α in mice with a heterozygous *Apc* gene mutation. The study noted that the use of BRB in an in vivo model led to a reduction in colon tumor volume and inhibition of intestinal epithelial proliferation. The study also proved that BRB can significantly reduce the pro-inflammatory activity of RAW 264.7 macrophages (decrease in the formation of the above-mentioned cytokines in vitro) in vitro and in vivo. Moreover, it was determined that BRB had an impact on the EGFR-ERK signaling pathway in colon cells both in vitro and in vivo. It has been shown that exposure to BRB reduced the phosphorylation of EGFR and ERK, which led to a silencing of this signaling pathway and thus limited the proliferation of CRC [77]. 

A summary of the anti-invasive activity of BRB against CRC cells is presented in Figure 2.

Deng et al. assessed the impact of prophylactic BRB therapy as a factor that restricts inflammation in the large intestine area, which can result in carcinogenesis related to inflammation. The results of the study correlated with reports from Yan et al. (2022) and Li et al. (2017) [78]. In their study, Dai et al. showed that BRB can induce the expression of the long non-coding RNA of CASC2 in CRC cells, which increases apoptosis of the mentioned cells. The expression of Bcl-2 (an anti-apoptotic molecule) was demonstrated to be decreased by BRB treatment. Using molecular analysis, it was shown that the long non-coding RNA of CASC2 binds AUF-1, which prevents its interaction with the mRNA of the Bcl-2 molecule and thus inhibits its formation in the active form in the CRC cell [79]. 

Huang et al. evaluated the effect of BRB and evodiamine on TGF-β-associated CRC tumorigenesis. The expression of DNMT 1,3A and 3B was increased by TGF-β, while BRB treated cells showed a decrease in DNMT expression and an increase in miRNA expression compared to cells that were exposed to TGF-β. The mechanism of anti-oncogenic activity of BRB and evodiamine was seen in the intensification of miRNA expression (molecules with documented anticancer activity), which in turn contributed to a decrease in the expression of DNA methyltransferases (DNMT) involved in the development of cancer [80]. In a different study, it was observed that BRB affects the growth and migration of CRC cells (line SW480 and HT-29) through a mechanism that is dependent on the function of the glucose-regulated protein 78 (GRP78). The study demonstrated concentration-dependent anticancer activity of BRB, with a noticeable effect already at low concentrations of the order of 20 μM. Using molecular analysis, it was shown that under the influence of BRB, there was a general decrease in the expression of GRP78 and a significant deficit in the surface-located isoform of the mentioned protein in the membrane. An increase in the frequency of apoptosis was observed among SW480 cells exposed to BRB (dependent on the concentration of 20 µM). BRB treatment resulted in a significant decrease in GRP78 expression, which led to inhibition of the expression of genes involved in CRC migration (*c-myc*, vimentin, and cytokeratin-encoding genes) and inhibition of the expression of anti-apoptotic proteins Bcl-xl and Bcl-2 [81].

The role of BRB in the process of reducing the development of CRC by inhibiting the influence of the insulin-like growth factor 2 mRNA-binding protein 3 (IGF2BP3) factor, which is significantly involved in the development of numerous cancers, was also examined. As in other studies, BRB has been shown to inhibit the proliferation of CRC cells (lines HCT116 and SW480) in vivo and in vitro. The Western blot analysis showed a significant increase in the expression of apoptosis-inducing factors, including cleaved caspase 3 and Bax. Using molecular analysis, a strong BRB downregulation of IGF2BP3 was detected. Importantly, inhibition of IGF2BP3 expression resulted in a decrease in active IGF2 mRNA translation in these cells, which in turn resulted in downregulation of the PI3K/AKT signaling pathway. At the cellular level, this resulted in inhibition of the cell cycle in the G0/G1 phase. The use of BRB in a new CRC therapy method is represented by these reports as a promising vision [82]. The use of BRB was also demonstrated in the field of Sonic Hedgehog pathway interference in CRC cells (lines HCT116 and SW480). BRB had a promising anticancer profile, leading to inhibition of CRC cell division at concentrations of ~27.6 µM (HCT116) and 29.14 µM (SW480). Proapoptotic activity of BRB against cells of both lines was also proved, which was defined as a decrease in the expression of Bcl-2 protein mRNA and an increase in Bax mRNA expression. The activation of caspase 3 and 9 in CRC cells was also observed after BRB exposure. The Sonic Hedgehog pathway was found to have significant attenuation (expressed as a decrease in cascade products) through molecular analysis of signaling cascades. This report is particularly important because overexpression of the Hedgehog pathway leads to the acquisition of cancer resistance to 5-fluorouracil, and BRB is a specific repressor of this process that can restore CRC sensitivity to this cytostatic drug [83]. Another study evaluated the effect of BRB on apoptosis and the decline in CRC cell migration. Molecular analysis assessed that the concentration-dependent apoptotic effect of BRB on CRC cells was most likely associated with increased expression of the long non-coding RNA of CASC2. By binding to the gene promoter of the mentioned protein, the molecule affected the signaling pathway of the Bcl-2 protein, which resulted in a decrease in the expression of the anti-apoptotic factor. It was also shown that the process of binding lnc-RNA to the promoter of the gene encoding the Bcl-2 protein requires the presence of EZH-2 methyltransferase, which is a factor critically limiting the possibility of this interaction in the event of its deficiency in the cell [79]. 

The Zhao et al. study showed that BRB can inhibit the development of CRC by inhibiting cancer steam-like (CSC) cells, which significantly reduce the aggressiveness of cancer (lines HCT116 and HT29). The mechanism by which BRB exhibits anti-oncogenic activity was analyzed by researchers through molecular analysis. It has been proven that the exposure of HCT116 and HT29 CSC cells to BRB leads to a decrease in m6A RNA modification and hyper-expression of proteins such as m6A demethylase and, among others, FTO (alpha-ketoglutarate-dependent dioxygenase) in vivo. The tumor invasiveness and growth inhibition in the in vivo model decreased due to the molecular changes described. The described mechanism allowed BRB to restore cancer cells’ sensitivity to chemotherapy using 5-fluorouracil and irinotecan, which is highly significant [84]. BRB has also been identified as a substance causing telomerase activity limitation and telomere erosion in HCT116 CRC cells. Importantly, the BRB of the other substances tested (silymarin, boldin) had the lowest IC50 (for telomerase inhibition) value, equal to 10.30 ± 0.89 µg/mL. At the level of the cell cycle, an increased percentage of cells retained in the G0/G1 phase and a decrease in the number of cells entering the G2/M phase was observed. BRB use had a significant impact on the expression of CCND1 and CDK4, which are important regulators of cell transition through cell cycle phases. The Western blot analysis showed a decrease in CDK4 concentration and an almost 100% increase in CCND1 synthesis [85].

A summary of the ability of BRB to block the cell cycle of CRC cells is shown in Figure 3.

The Wu et al. study examined how BRB affected the permeability of the CRC-affected colon cell membrane in relation to the potential reduction of cancer risk and inhibition of tumor progression. As in the studies mentioned above, BRB resulted in concentration-dependent inhibition (20.9–42.2% rate) of CRC growth in an in vivo murine model. ZO1 proteins and occludins were both upregulated when BRB was used, resulting in a general thickening of the mucous membrane and an increase in intestinal villi numbers. After the use of HPLC, a decrease in the concentration of polyamines (putrescine, spermidine) in CRC cells treated with BRB was detected with a simultaneous increase in the concentration of these compounds in the normal colon epithelium. BRB in cancer cells also inhibited the expression of ODC, HIF-1, and *c-myc* and stimulated the formation of OAZ1 and SSAT [86]. Another study assessed how BRB can affect the development of colon tumors in the course of FAP (murine model in vivo). The amount of colon adenomas was reduced by BRB at the tissue level compared to the control group. In addition, a decrease in COX-2 expression was observed, and thus a reduction in inflammation involved in the development of new and existing polyps What is particularly important is that BRB treatment led to polarization of the macrophage population from M2 to M1. An in vitro study showed that BRB-activated macrophages effectively and in a dose-dependent manner prevented cancer cells’ migration in vitro [87].

Shen et al. also demonstrated that BRB can inhibit the Sonic Hedgehog mediated signal cascade (SHH) in CRC cells (cell lines HT29 and SW480). Molecular analysis showed that BRB did not affect the RNA transcription process and most likely interfered with the translation of SHH-encoding mRNA. Interestingly, BRB inhibited the growth of CRC in vivo (murine model) and was not inferior to the cytostatic drug Vismodegib (GDC-0449). In macroscopic terms, blocking paracrine SHH activity in CRC stroma led to decreased VEGF release by the tumor, which resulted in downregulated angiogenesis in the tumor environment [88]. Othman et al. investigated the molecular basis of BRB action in CRC therapy. The study used BRB and BRB in complexes with Zn^2+^ ions and 5-fluorouracil. For Caco-2 CRC cells, the IC50 values for the compounds were as follows: 19.89 µg/mL, 10.49 μg/mL, 34.65 µg/mL. The combination of zinc and BRB led to accumulation of ROS (oxidative stress induction) in CRC at the cellular level, as well as an increase in p53, Bax, and caspase-3 expression with a simultaneous decrease in Bcl-2 protein formation. After exposure to BRB derivatives, a greater percentage of cells in the apoptosis process was observed [89]. BRB and *Andrographis paniculata* extract interact in a synergistic way that inhibits replication in the CRC cells of many different cell lines. In terms of cell cycle disorders, an increase in the number of cells in the G0/G1 phase was observed with a decrease in the number of cells in the G2/M phase. At the molecular level, inhibition of the expression of genes involved in the replication process was detected: FEN1, MCM7, PRIM1, MCM5, POLA1, MCM4, and PCNA. The impaired genes expression led to replication dysfunction and were associated with the cytostatic activity of both substances [90]. 

A summary of the pro-apoptotic activity of BRB against CRC cells is presented in Figure 4.

The ability of demethylenberberine to inhibit the progression of HCT-116 CRC line and induce apoptosis in these cells was also investigated. IC50 values for BRB and demethylenberberine were as follows: >40 µM and 5.365–7.229 µM. It has been proven that at the molecular level BRB and its derivative lead to suppression of the activity of the TGF-β and p-Smad2, p-Smad3 pathways. The result of this process was a decrease in the tumor’s ability to undergo epidermal–mesenchymal transition, which, in turn, decreased its aggressiveness. The silencing of Smad and TGF pathways was accompanied by increased cleaved caspase-9 expression. It was observed that exposure to BRB led to an increase in the percentage of CRC cells retained in the S phase of the cell cycle, which corresponds to cytostatic properties of the substance [91]. The Li et al. study evaluated the molecular effect of BRB on CRC in vitro and in vivo at the level of the cell signaling pathways lincROR and Wnt/β-catenin. In an in vitro model, suppression of both signaling pathways was described, which led to induction of CRC cell apoptosis. At the macroscopic level, inhibition of tumor growth was observed in an in vivo model (murine model) [92]. Ni et al. made a significant discovery regarding the ability of BRB to directly interfere with the expression of the *HEY2* gene, which is involved in the progression of CRC and its metastasis to the lungs and liver. Microarray analysis showed that BRB exposure significantly reduced the expression of signaling pathways associated with the development of CRC: Hippo, Rap 1, and Wnt. BRB in the mechanism of direct inhibition of expression (docking model) led to inhibition of hub-gene *HEY2* expression, which resulted in inhibition of mesenchymal–epithelial transformation and a decrease in the formation of E-cadherine, β-catenin, and cyclin D1 [93]. Further interesting reports on the molecular action of BRB in CRC cells come from the Yan et al. study. In vivo analysis showed that BRB inhibited CRC progression (in vivo murine model) and lactate and pyruvate production (in vitro) and induced apoptosis, which proved to be a profound disruption of colon tumor metabolism. What is more, it has been shown that BRB directly inhibits the enzyme PKM2 (pyruvate kinase M2) and stimulates its ubiquitination. Inactivating PKM2 results in a decrease in STAT3 factor phosphorylation, which in turn causes a decrease in Bcl-2 and cyclin D1 (anti-apoptotic factors) expression [94].

Another study showed that in vitro treatment of BRB with cell CRC lines HT-29, SW-480, and HCT-116 (IC50 was 34.6, 44.3, and 32.1 µM, respectively) led to a decrease in the expression of aquaporins 1, 3, and 5 in CRC cells and also increased the expression of the *PTEN* gene (phosphatase and tensin homolog deleted on chromosome 10). The increase in *PTEN* expression led to the silencing of the signaling pathways PI3K, AKT, and mTOR in cancer cells. This process led to an increase in the apoptosis level of CRC cells and a decrease in their migration ability [95]. A study was conducted to evaluate the influence of BRB on IGF-2 mRNA-binding protein-3 (IGF-2BP3), which is known to play a role in CRC development. Molecular analysis showed that BRB induces TRIM21-dependent ubiquitination of IGF-2BP3 protein. As a consequence of this process, the availability of IFG-2BP3 decreased, which made the CDK4 and CCND1 mRNA stabilization (cell division promoting factors) inefficient. CRC cells’ cell cycle was inhibited in the G1/S phase, which prevented the growth of cancer both in vivo and in vitro [96]. Ibrahim et al. described an interesting model of CRC therapy (pharmacologically induced tumor in an in vivo murine model) using BRB-containing liponiosomes. After BRB therapy, laboratory animals stopped losing body weight, and their diarrhea reduced and survival increased compared to the control group. The liponiosomal form of BRB was characterized (on a molecular level) by higher effectiveness than free BRB in the field of Bax, caspase-3 expression induction and inhibition of the expression of Bcl-2. The reduction of inflammation (downregulation of COX-2, IL-6, and TNF-α) also resulted in the same outcome. There was also an attenuation of the activity of the mTOR pathway and the NF-κB-mediated signaling cascade. Both processes resulted in the reduction of angiogenesis due to the suppression of vascular endothelial growth factor (VEGF) formation [97].

The Piao et al. study demonstrated that BRB has an impact on macrophage population polarization in a CRC environment caused by chronic colitis. BRB reduced the amount of inflammatory infiltrates of tumor tissue, while reducing the population of M2 macrophages in favor of M1. At the molecular level, BRB activity was associated with a decrease in micro-RNA-155-5p expression with a simultaneous increase in SOCS-1 cytokine modulator expression [98]. BRB can also be used to mobilize the immune system and stimulate cancer cell death by calreticulin expression (CRT) and CD47/SIRP-α interaction. The Mianowska et al. study assessed whether exposure of SW620 CRC cells to the liposomal form of BRB could lead to recruitment of an anti-inflammatory response from macrophages. At the molecular level, CRC cells exposed to the drug have been shown to develop oxidative stress (OXPHOS dysfunction, an increase in intracellular Ca^2+^ concentration). Significantly, liposomal BRB exposure led to increased expression of CRT molecules on the surface of the CRC cell membrane. It was also proved that the simultaneous use of liposomal BRB and anti-CD47 antibodies led to increased phagocytic activity of macrophages against CRC SW620 cells than in the case of both preparations separately [99].

### 2.2. Chelerythrine

Chelerythrine (CHL) and its derivatives have a potential anticancer effect. CHL chloride has been studied for its effects on CRC. The effect of this compound on inhibiting the G1 and S phases of the cell cycle, as well as increasing the amount of reactive oxygen species (ROS), has been demonstrated. α-SMA expression decreased, as determined by the examination of the effect on the regulation of two axes, WNT10B/β-catenin and TGFβ2/Smad2/3. α-SMA is a factor responsible for the production of cancer-associated fibroblasts, which play a role in the pathogenesis of metastasis [100]. The synthesis of C(6)-substituted dihydrobenzo[c]phenanthridines via copper-catalyzed C(sp3)-H functionalization of dihydrochelerythrine with nucleophiles to investigate the effect on cytotoxicity against CRC was also revealed. The compounds obtained were shown to have a cytotoxic effect on CRC (especially line HCT-8). Additionally, 6-acetonyldihydrochelerythrine, 6-cyanodihydrochelerythrine, and 6-methoxydihydrochelerythrine showed promising cytopathic effect on cancer cells (HCT-8, HCT116, and SW620). Dihydrochelerythrine derivatives obtained IC50 values compared to doxorubicin, ranging between 2.7–37.1 µg/mL for HCT-15 line cells [101]. Dihydrochelerythrine was obtained in the synthesis process, and then, after oxidation and acetonylation, the main product was (±)-6-acetonyldihydrochelerythrine. The activity of these compounds against cancer cells was tested and the effective action of (±)-6-acetonyldihydrochelerythrine against cells of the HCT 116 line was proved, while dihydrochelerythrine showed a weaker effect against cells of the A431 line. Dihydrochelerythrine and previously described similar compounds combined with DNA quadruplex promoters (*c-myc* and c-KIT1) could be implemented as an alternative to existing cancer therapies [102]. 

A summary of the anticancer properties of CHL is presented in Table 1.

### 2.3. Sanguinarine

Sanguinarine (SAN) has been demonstrated to have anticancer and antiproliferative effects on the HCT116 human CRC cell line, which is mediated through different apoptosis-related mechanisms. SAN-induced apoptotic signaling involves modulation of Bcl-2 and inhibitor of apoptosis protein (IAP) family proteins, mitochondrial disfunction, caspase activation, and induction of early growth response gene-1 (*Egr-1*). The study suggested that reactive oxygen species (ROS) are also a deciding factor in the SAN-induced apoptosis pathway. Exposing HCT-116 cells to 0.6 μM and 1.2 μM concentrations of SAN for 48 h led to inhibition rates of cell divisions of approximately 45% and 82%, respectively. The morphological changes observed under a microscope were indicative of apoptotic cell death, which involved cell shrinkage and extensive detachment from the cell culture substratum [103]. Gong et al. investigated the mechanisms regulating SAN-induced apoptosis in CRC cells. The study was conducted on SW-480 and HCT-116 CRC cell cultures and on orthotopic CRC models established on an in vivo murine model. Results of the study confirmed that SAN inhibited the growth of CRC cells by inducing apoptosis. The tumor weight of orthotopically implanted CRC models was significantly reduced in specimens treated with 4 mg/kg/day and 8 mg/kg/day of sanguinarine (administered via oral gavage). However, the body weight decrease was lower compared to that caused by cisplatin. CRC cell lines treated with SAN concentrations of 0.6, 1.2, 1.8, and 2.4 μM after observations lasting 24 h and 48 h showed apoptosis-characteristic features such as chromatin condensation, rounding up of the cells, and shrinkage and extensive detachment of the cells from the cell culture substratum in all concentrations. The cell viability was decreased by SAN in a manner that was both dose-dependent and time-dependent. Mechanisms of apoptosis were further studied, and results indicated that SAN shows effective antitumor activity through downregulating and dephosphorylating serine-threonine kinase receptor-associated protein (STRAP) and maternal embryonic leucine zipper kinase (MELK), which are potential pro-tumoral markers of CRC. SAN increased ROS levels and induced MOMP and disassociated the interaction between STRAP and MELK to trigger the intrinsic apoptosis pathway [104].

Another study was performed to explore oxeiptosis-mediated tumor suppression as a novel way to treat CRC using SAN regimens. CRC cell lines HT-29, HCT-116, Caco-2, and HT-115 were treated with SAN concentrations of 0.5, 1, 2, 3, 4, 5, 6, 7, and 8 μM. SAN was found to induce dose- and time-dependent cell death in all tested cell lines. The molecular mechanisms of cell death were further examined by performing next-generation sequencing (NGS) on the treated HT-29 cells. Multiple genes associated with different cell death pathways, including apoptosis, autophagy, necroptosis, ferroptosis, and oxeiptosis, showed differential modulation in HT-29 cells treated by SAN, indicating simultaneous activation of several programmed cell death pathways. Flow cytometry analysis showed characteristics indicative of both apoptosis and necroptosis in CaCo-2 and HT-29 cells. Elevated levels of intracellular ROS were observed, and intriguingly, ROS inhibitors such as glutathione (GSH) and N-acetyl-L-cysteine (NAC) significantly reduced SAN-related ROS production and features of apoptosis, necroptosis, oxeiptosis, autophagy, and cytotoxicity in CRC cells. To determine the role of oxeiptosis in SAN-induced tumor suppression in vivo, nude mice were inoculated with HT-29 cells and treated intraperitoneally with 6 mg/kg doses of SAN every 3 days for a period of 24 days. After 3 weeks of treatment, growth of the xenograft tumor was effectively suppressed. Additionally, indicators of oxeiptosis were found in the excised tumor tissue. To evaluate the toxicity of SAN, histopathological examination was conducted on the liver and kidney of mice, and there were no significant morphological changes between the samples treated with SAN and the control sample. This suggests that SAN effectively inhibited tumor growth in the HT-29 xenograft murine model, without causing additional toxicity to normal tissues such as liver and kidney [105].

## 3. Activity of Natural Compounds against Gastric Cancer

### 3.1. Berberine

Zhang et al. investigated the effect of BRB in vivo and in vitro on gastric cancer (GC) cells of the BGC-823 line (IC50 = 24.16 µM). The researchers showed that BRB induces the development of cytostatic autophagy of GC cells: 6 h after BRB application, the autophagy factor increased from 59% to 89%. The described phenomenon was accompanied by increased expression of Beclin-1, LC3-II, p-ULK1, and decreased expression of p62 (also described in in vivo analysis). At the molecular level, it was shown that autophagy induction was associated with inhibition of key signaling pathways: AKT, MAPK (ERK, JNK, p38), and mTOR/p70S6K. According to an in vivo study, mice exposed to BRB experienced a decrease in gastric tumor size (a decrease in tumor volume that ranges from 30 to 69.6% depending on BRB concentration) in vivo [106]. Significant reports on the synergistic interaction of BRB with 5-fluorouracil (5-FU) in GC therapy come from Pandey et al.’s study. The evaluation examined how BRB affects survivin expression and STAT3 signaling cascade activity, which could be responsible for cancer’s resistance to 5-FU treatment. At the molecular level, a decrease in the expression of survivin was detected after exposure of GC cells to BRB, and the issue of STAT3 activity (decrease in phosphorylation to p-STAT3) was similar. Both BRB (5 µM) and curcumin (10 µM) in combination with 5-FU have been shown to exhibit synergistic interaction in the field of STAT3 blockade and survivin expression. A similar correlation has been found in the field of inducing the expression of pro-apoptotic factors (cleaved form of caspase 3 and 9) [107]. 

Hu et al. evaluated the effect of BRB on the development and invasiveness of GC, with particular emphasis on molecular interference of BRB with expression of hepatocyte nuclear factor 1 homebox (*HNF-α*). In vivo and in vitro models demonstrated, as in previous studies, that BRB leads to impaired growth of gastric tumors as well as reduced invasion and cancer cells’ migration. Using in vitro analysis, it was observed that AGS and SGC7901 GC cells (after BRB administration) were characterized by reduced expression of *c-myc* and cyclin D1 (an increase in the percentage of cells retained in the G0/G1 phase of the cell cycle was observed). The decrease in MMP-3 expression led to a decrease in GC invasiveness. BRB has been confirmed to inhibit *HNF-α* gene expression with concurrently potentiating p-AMPK signaling. The decrease in *HNF* expression subsequently led to a reduction in expression of WNT-5A and β-catenin and to an increase in E-cadherin expression. In conclusion, due to the complex molecular mechanism, BRB led to a decrease in the aggressiveness and growth of GC in vivo and in vitro [108]. Potential synergistic interactions (in vivo and in vitro) between BRB with cetuximab and erlotinib therapy (both compounds have anti-EGFR activity) were also investigated. The ability of BRB to inhibit growth factor activation was associated with the decrease in EGFR phosphorylation observed in an in vitro study. In combination with cetuximab and erlotinib, there was a significant increase in the inhibition of SGC7901 and BGC823 GC cell lines. It has been proved that in both cases, there was a synergy of action between compounds (IC50 for BRB equal to 48 µM; erlotinib: 30 µM; after the combination of the substances, the activity of the latter increased 1.5 times). Blocking the EGFR signal caused the STAT3 pathway to be silenced and the expression of anti-apoptotic molecules Bcl-xl to decrease while the PARP marker increased. In addition, a decrease in the phosphorylation of EGFR, FGFR1, VEGFR2 was demonstrated [109].

The effect of BRB-based therapy on the apoptosis of GC cells of the BGC-823 and SGC7901 lines was also assessed. The apoptosis rate of the cells described and the PARP hyper-expression associated with it were validated by analyzing their concentration-dependent rate. Western blot molecular analysis showed a significant decrease in the phosphorylation of AKT to p-AKT after exposure of GC cells to BRB. The same was observed for the phosphorylation of AKT substrates mTOR, p70S6K, and S6. In the further part of the molecular analysis, it was proved that BRB interfered with the membrane potential of mitochondria and also contributed to the decrease in Bcl-2 expression. In vivo analysis (murine model) showed that BRB had a cytostatic potential similar to cyclophosphamide (46.58% vs. 48.86% inhibition of GC development) [110]. You et al. examined the ability of BRBs to restore cisplatin sensitivity in GC therapy on cytostatic drug-sensitive lines SGC7901 and BGC-823 and resistant mutants SGC-7901/DDP and BGC-823/DDP. According to the researchers, the use of BRB resulted in the induction of miR-203 expression in GC cells, which in turn inhibited the expression of the Bcl-w oncogene, leading to the induction of caspases 3 and 9 pathway (apoptosis). The process restored cisplatin sensitivity in a population of GC cells that were originally resistant to this drug. The presented data indicated the existence of a specific miR-203/Bcl-w axis, which determined the induction of apoptosis by BRB in gastric cancer cells [111]. Research on the potential synergy of BRB with cisplatin was also conducted by Kou et al. Molecular analysis showed that the combination of both substances improved the susceptibility of resistant mutants to cisplatin, which was most likely due to the silencing of the PI3K/AKT/mTOR signaling pathway. In addition, it was observed that exposure to BRB led to a decrease in the expression of drug-transporting proteins: MDR-associated protein 1, MDR1 [112]. The Li et al. study assessed whether the effect of the anticancer properties of BRB on MGC-803 GC cells was due to the interference of the substance with the MAPK signaling pathway. Both in vivo and in vitro analyses revealed that BRB suppressed MAPK (p38-MAPK, ERK1/2) and JNK phosphorylation rate and restricted IL-8 secretion in vivo and in vitro. The growth of gastric tumors was decreased by BRB as a result of these mechanisms [113]. 

BRB also induced dose-dependent inhibition of MGC-803 GC cell proliferation (IC50 = 40 µM). In GC cells, the mitochondrial apoptosis pathway was induced by a dose-dependent mechanism, with a decrease in Bcl-2 protein expression and an increase in Bax and caspase-3 protein expression. BRB also impaired lipid metabolism in MGC-803 cells, leading to a decrease in the use of both endogenous and exogenous triglycerides (accumulation of lipids in cancer cells). Fat metabolism disorder was identified with a BRB-mediated decrease in the expression of PPAR-α and FABP4/5. Furthermore, the GC cells experienced an increase in mitochondrial apoptosis due to the decreased FABP expression. Similar conclusions were drawn following in vivo analysis of the murine model [114]. There was also a significant relationship between the inhibition of development of GC and the use of BRB (20 µM) in combination with a low glucose diet (1.25 mM). Both in vitro (MGC803) and in vivo (xenograftmurine models) analysis have shown that the described therapy leads to activation of the signal axis PP2A/GSK3β. Activation of the described signaling pathways in turn led to a decrease in the expression of the anti-apoptotic factors MCL-1 and Bcl-2 with an increase in Bax expression, leading to the induction of GC cell apoptosis. The described combination of substances significantly reduced the migration of MGC803 cells, which is promising for the search for therapy to prevent the development of GC metastases [115].

A summary of the BRB impact on GC cells is presented in Figure 5.

The Yang et al. study showed that BRB inhibited the proliferation of GC cells of the SGC-7901 line and induced apoptosis in a concentration-dependent manner. The percentage of cells retained in the G1 phase also increased after BRB exposure. BRB has also been shown to induce the cellular formation of miRNA molecules that subsequently inhibit the expression of the mRNAs of key signaling proteins assigned to the following pathways: Hippo, Notch, FoxO, Ras, PI3K/Akt, and others. Analysis of miRNA–mRNA interactions allowed to detect new axes, which inhibited the development of GC [116]. BRB also contributed to the inhibition of the proliferation, migration, and invasiveness of MKN-45 (IC50 = 35.73 µg/mL) and HGC-27 (IC50 = 64.39 µg/mL) GC cells. In an in vivo model, a decrease in the size of gastric tumors was observed. The percentage of apoptotic GC cells increased significantly after BRB exposure, leading to the elevated expression of pro-apoptotic factors, such as Bax, and the decreased expression of Bcl-2 (anti-apoptotic molecule). Based on the evaluation of the expression of cyclin D1 (lowered) and p21 protein (increased), it was determined that cell cycle inhibition occurred in the G0/G1 phase. In addition, exposure to BRB (40 μg/mL) resulted in decreased expression of MMP-9 (matrix metaloproteinase-9); hence this exposure was associated with a decrease in the ability of GC cells to migrate. Finally, molecular analysis showed that BRB action results in a decrease in the expression of a number of IL-6/JAK2/STAT3 signaling pathway factors, thereby limiting GC proliferation dependent on the described pathway. STAT3 phosphorylation blockage led to the induction of cancer programmed cell death and an earlier arrest in the G0-G1 phase of the cell cycle [117].

The impact of BRB on GC growth in a murine xenograft (in vivo model) was investigated by Li et al., and it was also studied how BRB influenced the expression of the HNF4-WNT5a/-catenin signal pathway in cancer cells. In vivo treatment of BRB led to a reduction in GC line MGC803 and SGC7901 growth, of 50% and 60.9% (subcutaneous implantation in mice) and 48.6% and 51.3%, respectively, after xenograft implantation. BRB was able to inhibit tumor growth by reducing the expression of HNF4-α, WNT5a, and β-catenin in GC cells present in xenografts, as predicted [118]. The effect of chitosan/pectin nanoparticles loaded with BRB (NP-BRB) on the activity of AGS GC cells was also assessed. The nanoparticles themselves were characterized by higher cytotoxicity to GC cells than free BRB (IC50 = 63.44 μg/mL vs. 99.17 μg/mL for free BRB). The compound caused an increase in the percentage of cancer cells retained in the G0/G1 phase (the NP-BRB form had a more significant effect). BRB also resulted in increased expression of miR-185-5p, which is a known attenuator of the expression of the gene *KLF7*, encoding protein involved in GC metastasis and proliferation. GC cells experienced a decrease in tumor proliferation as a result of decreased DNMT1, 3A, and 3B expression. BRB was proven to be responsible for the hypermethylation of cytidine DNA residues, resulting in a reduction in GC chromosomal instability and tumor malignancy [119].

### 3.2. Chelerythrine

There are also a few reports of the anticancer properties of CHL against GC. CHL acts on GC cell lines (NCI-N87) through time and dose dependence. However, the most important function is the inhibition of thioredoxin reductase oriented to Sec 498 (TXNRD1). This was assessed in the DTNB-reducing activity assay, in which the IC50 of this compound reached a value of 65.9 μM. This compound has additional beneficial effects that are linked to TXNRD1 inhibition and involve the production of ROS and the induction of oxidative stress on the cell organelle known as the endoplasmic reticulum. Each of these actions led to the death of the cancer cell [120]. In another study, CHL was evaluated for its impact on the PI3K/AKT pathway, and it was concluded that the compound inhibits this pathway. What is more, the expression of the PI3Kca protein was reduced due to CHL exposure. Moreover, it also reduced the growth of cancer cells and their migration, leading to cell death [121]. An additional study on CHL demonstrated that it had a significant impact on the expression of COX-2 and 5-lipooxygenase (5-LOX), inhibiting the activity of these enzymes. The saturation values for COX-2 and 5-LOX were 7.81 and 24.49, respectively, showing a time- and dose-dependent relationship. The effect of CHL on COX-2 and 5-LOX suggested by the authors of the study may be related to cross-reactions to the pathways of thyroid hormones, estrogens, and oxytocin [122]. The effect of CHL on Janus kinases, which participate in the process of cancer development, was also demonstrated. CHL showed higher selectivity towards Janus kinases JAK 1,2,3 than towards TYK2. Additionally, it caused induction of apoptosis and inhibition of gastric cancer cell migration and their adhesion ability. The study showed a relationship of CHL with a reduction in the number of estrogen receptors in the cancer cell membranes [123]. 

**Table 1 ijms-25-08375-t001:** Anticancer activity of CHL against CRC and GC cells.

Anticancer Activity of Chelerythrine		
Type of Cancer	Molecular Effect of Action	References
CRC	Cell cycle inhibition in G1/S phase Oxidative stress induction WNT10B/β-catenin and TGFβ2/Smad2/3 pathways inhibition	[100]
CRC (HCT-8, HCT-15, HCT-116, SW620)	Apoptosis induction	[101]
CRC (HCT-116, A431)	Direct binding with promoters of genes: *c-myc, c-KIT1*	[102]
GC (NCI-N87)	Inhibition of TXNRD1 Oxidative stress induction Apoptosis induction	[120]
GC	PI3K/AKT signaling pathway silencing due to PI3Kca downregulation	[121]
GC	Blockage of inflammation-induced carcinogenesis due to inhibition of COX-2 and 5-LOX expression and activity	[122]
GC	JAK/TYK pathway inhibition-related induction of apoptosis Estrogen receptors downregulation	[123]

### 3.3. Sanguinarine 

Zhang et al. analyzed the correlation between expression of dual-specificity phosphatase 4 (DUSP4) and survival in patients with GC and also investigated the effects of SAN on tumor growth and invasion in GC cell lines SGC-7901 and HGC-27, along with the underlying molecular mechanisms. DUSP4 is a member of the phosphatase family and plays an important role in physiological and pathological cell processes such as cell growth, apoptosis, and carcinogenesis. Loss of DUSP4 in genome is a common occurrence in various cancer types, which suggests that DUSP4 may function as a tumor suppressor. In the conducted study, SAN concentrations of 5, 10, and 30 μM were shown to exhibit an inhibitory effect on GC cell growth, decreasing cell proliferation in a dose- and time-dependent manner while having minimal inhibitory impact on GES-1 (gastric epithelial cell line) cells. Additionally, GC cell invasion was reduced in a dose-dependent manner. During subsequent investigations on the impact of SAN on GC cells, it was discovered that it also led to cycle arrest in the S phase and induction of apoptosis in GC cells, while DUSP4 expression was increased by SAN in a dose-dependent way [124]. One of the key factors in the pathological process of various malignancies is dysregulation in the signaling pathways of DNA repair, such as the nonhomologous end joining (NHEJ) pathway. DNA-dependent protein kinase (DNA-PK), formed by its catalytic subunits (DNA-PKcs) bound with the KU70/80 heterodimer, is a key promoter of NHEJ. TOX is a protein that binds to DNA and plays a role in regulating apoptosis and DNA repair. Abnormal expression of TOX determines tumor growth by binding with KU70/80 and inhibiting NHEJ repair. According to a study conducted by Fan et al., low TOX expression is associated with poor survival in patients with GC. Fan et al. studied the effects of SAN on tumorigenesis of GC and on TOX/DNA-PKcs/KU70/80 signaling. GC cell lines SGC-7901 and AGS were exposed to SAN concentrations of 0 μM, 1.25 μM, 2.5 μM, and 5 μM, while nude mice with xenograft tumor models were treated with 4 mg/kg and 8 mg/kg of SAN. The findings revealed that the concentrations of SAN used were able to inhibit the growth of GC cells and the formation of colonies, causing cell cycle arrest and apoptosis and inhibiting xenograft tumor growth. Exposure to SAN also caused a considerable increase in TOX expression levels in GC cells, but significantly reduced the expression of downstream DNA-PKcs and KU70/80 [125]. A different molecular mechanism of SAN-induced inhibition of GC cell proliferation was studied by Dong et al. According to their study, SAN’s antitumor function in BGC-823 GC cells was influenced by the expression of miRNAs miR-96-5p and miR-29c-3p (miRNAs) and the mitogen-activated protein kinase (MAPK) signaling pathway, whose activation is associated with cell cycle arrest and apoptosis induction. In the study, human GC cells of MGC-803, BGC-823, and SGC-7091 cell lines were incubated with SAN concentrations of 0, 50, 100, 200, 300, and 400 μM, while nude mice with a xenotransplanted tumor model (established from the BGC-823 cell line) were treated with 2.5 mg/kg, 5 mg/kg, and 10 mg/kg SAN (through intraperitoneal administration). SAN caused a concentration-dependent reduction in the viability of GC cells both in vitro and in vivo [126]. A summary of the anticancer properties of SAN is presented in Table 2.

**Table 2 ijms-25-08375-t002:** Anticancer activity of SAN against CRC and GC cells.

Anticancer Activity of Sanguinarine		
Type of Cancer	Molecular Effect of Action	References
CRC (HCT-116)	Oxidative stress induction Induction of mitochondrial apoptosis pathway (Bcl-2 ↓, IAP ↓, caspase 3 and 9 ↑) *Egr-1* gene expression ↑	[103]
CRC (SW480, HCT-116)	Downregulation and dephosphorylation of STRAP and MELK Oxidative stress induction MOMP ↑	[104]
CRC (HT-29, HCT-116, CaCo-2, HT-115)	Oxidative stress induction Activation of various programmed cell death pathways	[105]
GC (SGC-7901, HGC-27)	Cell cycle arrest in S phase Apoptosis induction Upregulation of DUSP4	[124]
GC (SGC-7901, AGS)	Interference with signal pathway TOX/DNA-PKcs/Ku70/80 Cycle cell arrest, apoptosis pathway induction	[125]
GC (MGC-803, BGC-823, SGC-7091)	miRNA and MAPK pathway-induced cell cycle arrest and apoptosis	[126]

## 4. Materials and Methods

In this review, we searched for articles by using the databases Scopus, PubMed, Web of Science, and Google Scholar. In total, 125 articles were cited. The articles were qualified for review by searching for the following keywords in the title and abstract of the articles: ‘‘colorectal cancer’’, ‘‘gastric cancer’’, ‘‘berberine’’, ‘‘chelerythrine’’, and ‘‘sanguinarine’’. The years of publication of the qualified articles were reduced after 2012. Figure 1 shows the chemical formulas of alkaloids described in the review, and Figure 2, Figure 3, Figure 4 and Figure 5 depict simplified schemes of the molecular action of berberine on cancer cells. The meanings of arrow pictograms are: ↓ (downregulation) and ↑ (upregulation). Table 1 and Table 2 present a summary of the spectrum of the molecular effects of chelerythrine and sanguinarine against both cancers. 

## 5. Conclusions

Compounds of natural origin are an interesting group of substances with a broad spectrum of activity, and their action can be successfully implemented in medicine. So far, many of the widely used cytostatic drugs have been isolated from plants: I—vinblastine and vincristine (alkaloids of *Catharanthus roseus* Linn); II—topotecan and irinotecan (derivatives of camptothecin—*Camptotheca acuminate* Decne); III—paclitaxel and docetaxel (taxanes derived from *Taxus brevifolia* Nutt); and IV—colchicine (an alkaloid from the *Colchicum autumnale* L.). Indeed, the fact that numerous drugs with anticancer activity are essentially alkaloids of plant origin is an important rationale for the search for more such substances in the plant reservoir. The organic compounds described in this article fit this definition. 

The summarization of available reports on the activity of berberine, chelerythrine, and sanguinarine in treating CRC and GC, the most common GIST cancers, was accomplished by us. Conclusions from the studies we cited unequivocally indicate that these compounds are capable of (i) inducing apoptosis of cancer cells, (ii) inhibiting the cell cycle, and (iii) downregulating the expression of crucial oncogenes for tumorigenesis. At the molecular level, all of the reported compounds led to the interference of critically important signaling pathway cascades for carcinogenesis. Reports of BRB re-establishing sensitivity to platinum-derivative alkylating agents in resistant GIST cancers seem to be particularly significant. By breaking the intrinsic resistance of the tumor, BRB also induced a synergistic effect against antimetabolite cytostatic agents (5-FU) and topoisomerase-I inhibitors (irinotecan). The described alkaloids’ pleiotropic mechanism of action offers an intriguing alternative for the development of oncological therapy that can be combined with conventional cytostatic drugs. Products of natural origin could, in such a combination, act as agents that increase the effectiveness of anticancer drugs, as well as protect against the development of resistance generated by the tumor during treatment (e.g., overexpression of survivin). Considering the cited facts, preparations of naturally derived substances such as berberine, chelerythrine, and sanguinarine appear as a promising group of substances worthy of deeper investigation, in terms of cancer therapy.

## 6. Abbreviation List in Alphabetical Order

5-FU—5-fluorouracil, 5-LOX—5-lipooxygenase, AKT—protein kinase B, Apc—tumor suppressor gene, adenomatous polyposis coli, AUF-1—AU-rich element RNA-binding protein 1, Bax—apoptosis regulator BAX, Bcl-2—B-cell lymphoma 2 (regulator protein), Bcl-w—Bcl-2-like protein 2, Bcl-xl—B-cell lymphoma extra-large, Beclin-1—regulator of autophagy and cell death, BRAF—v-Raf murine sarcoma viral oncogene homolog B, BRB—berberine, CCND1—cyclin D1, CD—cluster of differentiation, CDK4—cyclin-dependent kinase 4, c-KIT1—proto-oncogene c-KIT1, c-myc—cellular myc regulator gene, CHL—chelerythrine, COX-2—cyclooxygenase 2, CRC—colorectal cancer, CRT—calreticulin, CSC—cancer stem cells, CTLA-4—cytotoxic T-lymphocyte associated protein 4, DNA-PK—DNA-dependent protein kinase, DNMT—DNA methyltransferase, DUSP4—dual-specificity protein phosphatase 4, EGFR—epidermal growth factor receptor, Egr-1—early growth response protein 1, ERK—extracellular signal-regulated kinase, ESβL—extended spectrum β-lactamase, EZH-2—enhancer of zeste homolog 2, FABP4—fatty acid-binding protein 4, FAP—familial adenomatous polyposis, FEN1—flap structure-specific endonuclease 1, FGFR1—fibroblast growth factor receptor 1, FoxO—forkhead box O, FTO—alpha-ketoglutarate-dependent dioxygenase FTO, GC—gastric cancer, GIST—gastrointestinal stromal tumors, GRP78—glucose-related protein 78, GSK3β—glycogen synthase kinase-3 beta, HCC—hepatocellular carcinoma, HEY2—hairy/enhancer-of-split related with YRPW motif protein 2, HIF-1—hypoxia induced factor 1, Hippo—Salvador–Warts–Hippo pathway, HNF-α—hepatocyte nuclear factor 1 homeobox A, HPLC—high-performance liquid chromatography, Hsp90—heat shock protein 90, IC50—half maximal inhibitory concentration, IGF-2—insulin-like growth factor, IGF2BP3—insulin-like growth factor 2 binding protein 3, IL—interleukin, JAK—Janus-activated kinase, JNK—c-Jun N-terminal kinases, Ki-67—antigen Kiel 67, KIT/PDGFRA—tyrosine-protein kinase KIT/platelet-derived growth factor receptor A, KLF7—Kruppel-like factor 7, KU70/80—Ku heterodimer protein, LC3-II—microtubule-associated protein 1A/1B-light chain 3 phosphatidylethanolamine conjugate, lincROR—long intergenic non-coding RNA, regulator of reprogramming, lnc-RNA CASC-2—long non-coding RNA of cancer susceptibility candidate 2 protein, MAPK—mitogen-activated protein kinase, MCL-1—induced myeloid leukemia cell differentiation protein Mcl-1, MCM4—DNA replication licensing factor MCM4, MCM5—DNA replication licensing factor MCM5, MCM7—DNA replication licensing factor MCM7, MDR1—multidrug-resistance protein 1 (P-glycoprotein 1), MELK—maternal embryonic leucine zipper kinase, miR-micro RNA, MMP—matrix metalloproteinase, MOMP—mitochondria outer membrane permeability, MRSA—methicillin-resistant S. aureus, mTOR—mammalian target of rapamycin, NF-κB—nuclear factor κB, NHEJ—non-homologous end joining pathway, Notch—neurogenic locus notch homolog protein, NP—nanoparticle, NSCLC—non-small cell lung cancer, OAZ1—ornithine decarboxylase antizyme, ODC—ornithine decarboxylase, OXPHOS—oxidative phosphorylation, p21—cyclin-dependent kinase inhibitor 1, p38—p38 mitogen-activated protein kinase, p53—tumor protein P53, p65—transcription factor p65, p70S6K—ribosomal protein S6 kinase beta-1, Par3—partitioning defective protein 3, PARP—poly [ADP-ribose] polymerase 1, PCNA—proliferating cell nuclear antigen, PD-1—programmed cell death protein 1, PGE2—prostaglandin E2, PI3K—phosphoinositide 3-kinase, Pin1—peptidyl-prolyl cis/trans isomerase, PKM2—pyruvate kinase M2, POLA1—DNA polymerase alpha catalytic subunit, PP2A—protein phosphatase 2, PPAR-α—peroxisome proliferator-activated receptor α, PRIM1—DNA primase small subunit, PTEN—phosphatase and tensin homolog, p-ULK1—autophagy-related kinase, RAF—rapidly accelerated fibrosarcoma kinase, Rap1—Ras-related protein 1, RAS—rat sarcoma virus oncogene, ROS—reactive oxygen species, SAN—sanguinarine, SCFA—short-chain fatty acid, SHH—Sonic Hedgehog, SIRP-α—signal-regulatory protein α, Smad—mothers against decapentaplegic homolog protein, SOCS1—suppressor of cytokine signaling 1, SSAT—spermidine/spermine-N(1)-acetyltransferase, STAT3—signal transducer and activator of transcription kinase, STRAP—serine–threonine kinase receptor-associated protein, TGF-β—transforming growth factor β, TLR4—toll-like receptor 4, TNF-α—tumor necrosis factor α, TOX—thymocyte selection-associated high mobility group box protein TOX, TRIM21—E3 ubiquitin-protein ligase TRIM21, TXNRD1—cytoplasmic thioredoxin reductase 1, TYK2—non-receptor tyrosine-protein kinase, VEGF—vascular endothelial growth factor, VEGFR2—vascular endothelial growth factor receptor 2, Wnt/WNT—Wnt signaling pathway, ZO1—zonula occludens-1.

## Figures and Tables

**Figure 1 ijms-25-08375-f001:**
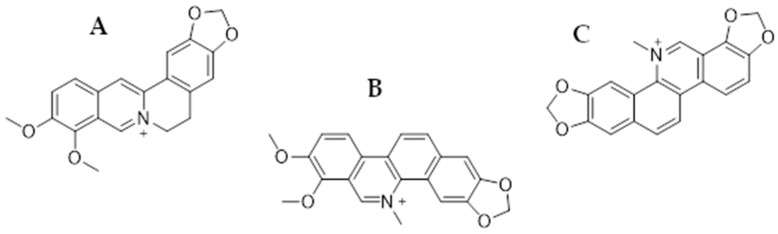
Chemical structures of berberine (**A**), chelerythrine (**B**), and sanguinarine (**C**).

**Figure 2 ijms-25-08375-f002:**
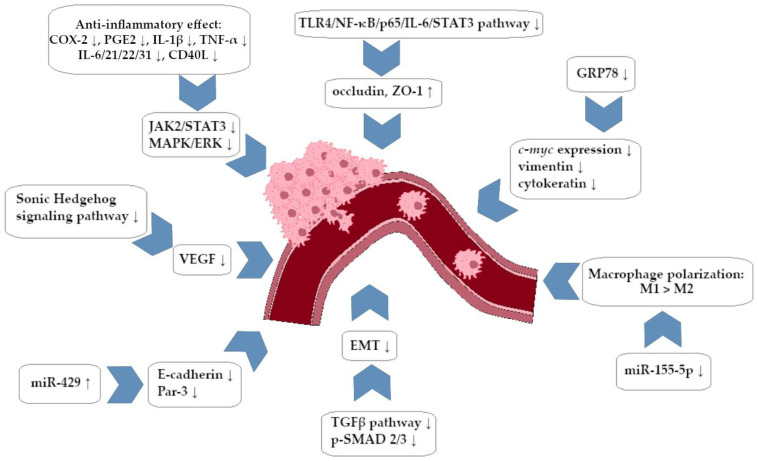
Berberine-dependent anti-invasive impact on colorectal cancer cells (simplified scheme).

**Figure 3 ijms-25-08375-f003:**
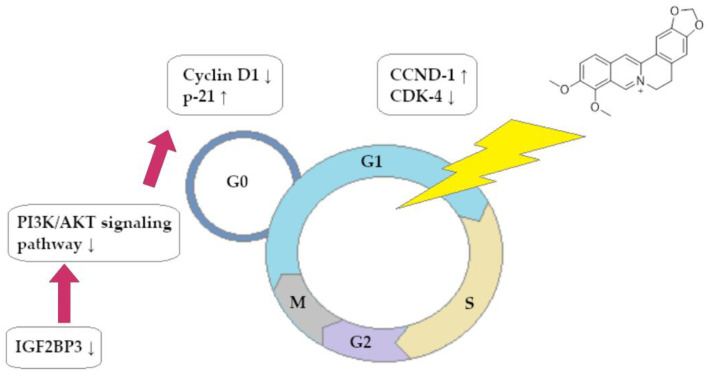
Berberine-dependent inhibition of cell cycle of colorectal cancer cells in phase G0/G1.

**Figure 4 ijms-25-08375-f004:**
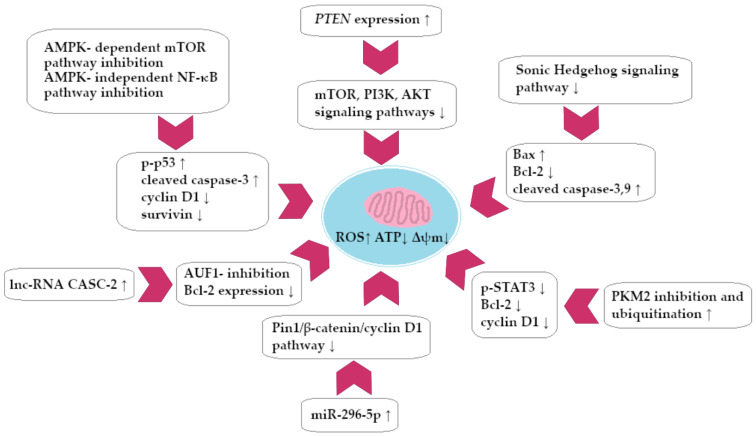
Berberine-dependent colorectal cancer cell apoptosis (simplified scheme).

**Figure 5 ijms-25-08375-f005:**
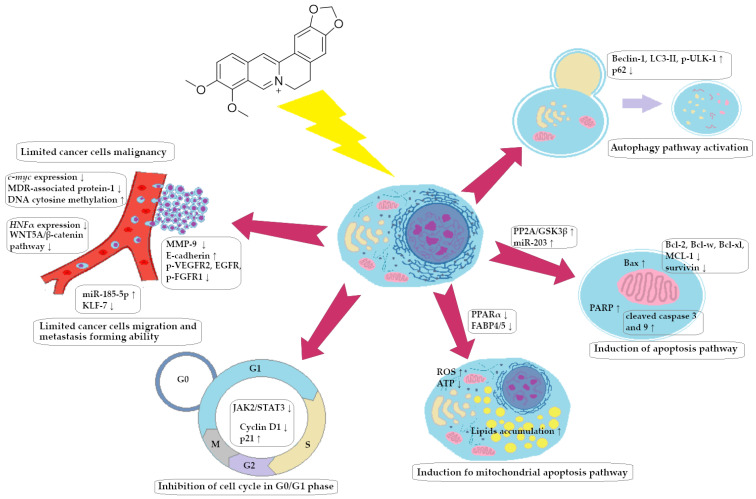
Spectrum of molecular effects of berberine on gastric cancer cells.

## Data Availability

Not applicable.

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
