# Peer review of "Natural Alkaloids in Cancer Therapy: Berberine, Sanguinarine and Chelerythrine against Colorectal and Gastric Cancer"

_ijms, 2024, doi:10.3390/ijms25158375_

Round 1

Reviewer 1 Report

Comments and Suggestions for Authors

In the review: "Natural alkaloids in cancer therapy: berberine, sanguinarine 2 and chelerythrine against colon and stomach cancer", the authors summarized the anticancer properties of berberine, sanguinarine and chelerythrine. Discovering new substances with anti-cancer activity is always of great importance. Especially with such a detailed description of the molecular properties of the substance provided by the authors of the review.

I wonder why the authors chose these three alkaloids.

Were all reported results regarding overall antitumor activity based on in vitro studies or on in vivo studies? If so, it would be worth highlighting it, separating it in the text or briefly presenting it in a table. I understand that determining the molecular mechanisms of action of compounds takes place mainly on cell lines, but then research, among others, on animals, provide additional, important information.

Line 126/127 - It also has antimicrobial activity against drug-resistant strains, e.g. MRSA or ESβL. - Information about strains is provided - ESBL. Does it apply to all ESBL-producing bacteria or to specific ESBL-producing bacteria? Authors think about: Escherichia coli (E. coli), Klebsiella pneumoniae, Pseudomonas aeruginosa, K. oxytoca, Proteus mirabilis, Salmonella enterica, Neisseria gonorrhoeae, Haemophilus influenzae.

Line 154 – there is PGE2, in the 156 line there is PGE-2. PGE2 should be standardized throughout the work. This is PGE2.

Line 157-158 - A multicenter, double-blind, randomized clinical trial evaluated the ability of BRBs  to prevent recurrence of colorectal adenoma (a factor that promotes the development of CRC). - In my opinion, the information in brackets is unclear whether it means that BRB inhibits a specific factor.

Line 190-191: “The authors reported that the decoction led to a significant decrease in the concentration of pro-inflammatory cytokines, including. IL-1β, TNF-α, COX-2.” - Something is wrong with these sentences, something is missing. COX-2 is not a cytokine.

On the figure 2:

Maybe it would be better to put the arrow in a different place, because one may get the impression that it only applies to TNF and CD40L.

Line 460: mainwas or main was

Author Response

Dear Reviewer

Thank you very much for the time you took to read our Manuscript. Thank you for your valuable comments, which we have taken into account as follows:

I wonder why the authors chose these three alkaloids.

Thank you very much for this question. In the Introduction, we mentioned that the lack of highly effective chemotherapy for malignant tumors is causing researchers to search for new solutions, which could involve using natural substances to treat cancer. The three alkaloids described in the review fit into this definition, which is why we decided to include them in this publication. 

In addition, we would like to inform the reviewer that among the many alkaloids we tested, it was these three that showed the highest antimicrobial activity (data not yet published). Therefore, we were interested in the antitumor activity of just these particular three compounds. Analyzing the results during the writing of this Review, it turned out again that they constitute a very select, complementary “ trio”. This will provide us with a basis for planning experiments in the laboratory.

Were all reported results regarding overall antitumor activity based on in vitro studies or on in vivo studies? If so, it would be worth highlighting it, separating it in the text or briefly presenting it in a table. I understand that determining the molecular mechanisms of action of compounds takes place mainly on cell lines, but then research, among others, on animals, provide additional, important information.

As it was mentioned in the introduction of our review, we focused on describing mainly molecular basis of action of the three alkaloids against GC and CRC cells. The majority of papers cited revealed the activity of those compounds both in vivo, in vitro so we decided not to separate the reports from cited articles but to present a comprehensive view on the activity of tested alkaloids. In our opinion it resulted in composited, profund view on the impact of substances on the e.g. signaling pathways in cancer cells. At first we thought about separating the in vivo and in vitro studies but during the process of writing the draft, it became clear that it will end in excessive long review written in repetitive manner, because both studies revealed the same topics and many times tested the same molecular aspect of carcinogenesis. Moreover because of level of complexity of molecular action of mentioned alkaloids in cancer therapy, we decided to add many colour pictures to make it easier for the reader to understand the relationship between alkaloids action on specific stage of cancer.

Line 126/127 - It also has antimicrobial activity against drug-resistant strains, e.g. MRSA or ESβL. - Information about strains is provided - ESBL. Does it apply to all ESBL-producing bacteria or to specific ESBL-producing bacteria? Authors think about: Escherichia coli (E. coli), Klebsiella pneumoniae, Pseudomonas aeruginosa, K. oxytoca, Proteus mirabilis, Salmonella enterica, Neisseria gonorrhoeae, Haemophilus influenzae.

Our aim was to highlight that the alkaloid's activity was tested against various ESBL producers, and the general property of this substance was to overcome this resistance mechanism, as discussed in the literature research we conducted.

Line 154 – there is PGE2, in the 156 line there is PGE-2. PGE2 should be standardized throughout the work. This is PGE2.

Thank you for your attention. We have replaced all PGE-2 with PGE2.

Line 157-158 - A multicenter, double-blind, randomized clinical trial evaluated the ability of BRBs  to prevent recurrence of colorectal adenoma (a factor that promotes the development of CRC). - In my opinion, the information in brackets is unclear whether it means that BRB inhibits a specific factor.

Thank you for your attention. We have corrected the sentence mentioned and we hope that the reader is able to understand our intention.

Line 190-191: “The authors reported that the decoction led to a significant decrease in the concentration of pro-inflammatory cytokines, including. IL-1β, TNF-α, COX-2.” - Something is wrong with these sentences, something is missing. COX-2 is not a cytokine.

We are grateful for your attention and have made a correction to express that we have taken into account the decreased activity of COX-2 enzyme.

Maybe it would be better to put the arrow in a different place, because one may get the impression that it only applies to TNF and CD40L.

Thank you for bringing this to our attention. To emphasize the down-regulation of every cytokine and enzyme mentioned, we added some extra arrows. We hope that our figures are now clearer and do not confuse the reader.

Line 460: mainwas or main was

Thank you for your attention. We have added the necessary space.

I hope that the corrections we have made are satisfactory and improve the quality of the Manuscript.

Kind regards,

Anna Duda-Madej

Reviewer 2 Report

Comments and Suggestions for Authors

Dear Editor

Many thanks for considering me as a potential reviewer for the article "Natural Alkaloids in Cancer Therapy: Berberine, Sanguinarine and Chelerythrine against Colorectal and Gastric Cancer". No doubt the article is well-structured, presented and well-written. However, I have several observations, which I believe should be taken into consideration before proceeding further.

My observations are as follows.

My observations

·       Introduction; a) Generally, the introduction is very poorly cited please pay attention to each claim and cite it accordingly, b), Line-59, What do you mean? ‘It is also interesting to note that both cancers discussed in this review share a common characteristic regarding their incidence about geographic location’.

·        Why much attention is given to Berberine than the other two?

·       Figures quality is not off the standard and must be improved,  

·       Line-16, please italicize In vivo and in vitro and do the said throughout MS,

·       All scientific names i.e. plants should be in italics and also mention authority names, accordingly.

·       Line-653, Several studies have also been conducted that describe the effect of CHL or its derivatives on gastric cancer cells. Where is several citations?

·       3.2. Chelerythrine wrote in two paragraphs and you just cited three articles? In a similar way, 3.3. Sanguinarine is described in three paragraphs and you had just cited three articles?, very strange.  

·       If possible, please put a table summarizing different biological activities of the said alkaloids in different forms of cancer and/or cancer cell lines.  

·       Why only 110 references were these important?

Comments on the Quality of English Language

Dear Editor/Authors,

I read the article but I am not happy with the quality of the English used. Several issues have been observed, and based on that I will highly recommend this manuscript for extensive English editing by a Professional and/or senior professor in the field. 

Author Response

Dear Reviewer

Thank you very much for the time you took to read our Manuscript. Thank you for your valuable comments, which we have taken into account as follows:

Introduction; a) Generally, the introduction is very poorly cited please pay attention to each claim and cite it accordingly,

Thank you very much for your attention. We have carefully revised the Introduction and added the following literature items where we thought necessary:

  1. Sbaraglia, M.; Businello, G.; Bellan, E.; Fassan, M.; Dei Tos, A.P. Mesenchymal Tumours of the Gastrointestinal Tract. Pathologica 2021, 113, 230–251, doi:10.32074/1591-951X-309.
  2. Inoue, A.; Ota, S.; Yamasaki, M.; Batsaikhan, B.; Furukawa, A.; Watanabe, Y. Gastrointestinal Stromal Tumors: A Comprehensive Radiological Review. Jpn. J. Radiol. 2022, 40, 1105–1120, doi:10.1007/s11604-022-01305-x.
  3. Gama, J.M.; Oliveira, R.C. Mesenchymal Tumors of the Gastrointestinal Tract—Beyond GIST—A Review. Gastrointest. Disord. 2024, 6, 257–291, doi:doi.org/10.3390/gidisord6010019.
  4. Siegel, R.L.; Wagle, N.S.; Cercek, A.; Smith, R.A.; Jemal, A. Colorectal Cancer Statistics, 2023. CA. Cancer J. Clin. 2023, 73, 233–254, doi:10.3322/caac.21772.
  5. Wang, S.; Yuan, Z.; Ni, K.; Zhan, Y.; Zhao, X.; Liu, Z.; Liu, Y.; Yi, B.; Lai, S.; Yin, X.; et al. Young Patients With Colorectal Cancer Have Higher Early Mortality but Better Long-Term Survival. Clin. Transl. Gastroenterol. 2022, 13, e00543, doi:10.14309/ctg.0000000000000543.
  6. Siegel, R.L.; Jakubowski, C.D.; Fedewa, S.A.; Davis, A. Colorectal Cancer in the Young : Epidemiology , Prevention , Management. 2020, 75–88.
  7. Majek, O.; Gondos, A.; Jansen, L.; Emrich, K.; Holleczek, B.; Katalinic, A.; Nennecke, A.; Eberle, A.; Brenner, H. Sex Differences in Colorectal Cancer Survival: Population-Based Analysis of 164,996 Colorectal Cancer Patients in Germany. PLoS One 2013, 8, 1–7, doi:10.1371/journal.pone.0068077.
  8. Andrilla, C.H.A.; Moore, T.E.; Man Wong, K.; Evans, D. V. Investigating the Impact of Geographic Location on Colorectal Cancer Stage at Diagnosis: A National Study of the SEER Cancer Registry. J. Rural Heal. 2020, 36, 316–325, doi:10.1111/jrh.12392.
  9. Torabi, M.; Green, C.; Nugent, Z.; Mahmud, S.M.; Demers, A.A.; Griffith, J.; Singh, H. Geographical Variation and Factors Associated with Colorectal Cancer Mortality in a Universal Health Care System. Can. J. Gastroenterol. Hepatol. 2014, 28, 191–197, doi:10.1155/2014/707420.
  10. Zhang, S.; Xiao, X.; Yi, Y.; Wang, X.; Zhu, L.; Shen, Y.; Lin, D.; Wu, C. Tumor Initiation and Early Tumorigenesis: Molecular Mechanisms and Interventional Targets. Signal Transduct. Target. Ther. 2024, 9, 1–36, doi:10.1038/s41392-024-01848-7.
  11. Joyce, C.; Rayi, A.; Kasi, A. Tumor-Suppressor Genes; StatPearls: Treasure Island (FL): StatPearls Publishing, 2023;
  12. E.E., V.; E., C. Anti-EGFR Therapes: Clinical Experience in Colorectal, Lung, and Head and Neck Cancers. Onkology 2006, 20, 15–25.
  13. Proietti, I.; Skroza, N.; Michelini, S.; Mambrin, A.; Balduzzi, V.; Bernardini, N.; Marchesiello, A.; Tolino, E.; Volpe, S.; Maddalena, P.; et al. BRAF Inhibitors: Molecular Targeting and Immunomodulatory Actions. Cancers (Basel). 2020, 12, 1–13, doi:10.3390/cancers12071823.
  14. Bauer, S.; George, S.; von Mehren, M.; Heinrich, M.C. Early and Next-Generation KIT/PDGFRA Kinase Inhibitors and the Future of Treatment for Advanced Gastrointestinal Stromal Tumor. Front. Oncol. 2021, 11, 1–11, doi:10.3389/fonc.2021.672500.
  15. Liu, J.; Chen, Z.; Li, Y.; Zhao, W.; Wu, J.B.; Zhang, Z. PD-1/PD-L1 Checkpoint Inhibitors in Tumor Immunotherapy. Front. Pharmacol. 2021, 12, 1–8, doi:10.3389/fphar.2021.731798.

 b), Line-59, What do you mean? ‘It is also interesting to note that both cancers discussed in this review share a common characteristic regarding their incidence about geographic location’.

Thank you for pointing out our not-so-clear wording. We meant that according to the WCRFI incidence data (per population) of GIST, most of the countries of occurrence of GC and CRC are overlapping. It is therefore speculated that geographic coordinates may influence the incidence of specific types of cancer (GIST is mentioned here). Our not-so-clear sentence has been changed to the following:

„It is also interesting to note that the geographic location of the two cancers discussed in this review is their common feature [17,18]. Many of the countries with the highest recurrence rates are overlapping for both GC and CRC”

We hope it is no more doubtful in its current form.

  • Why much attention is given to Berberine than the other two?

The berberine was the main contributor to our review volume, as it anticancer activity was well-documented and many articles were found in this field. Unfortunately, two other alkaloids were poorly revealed, and we found only a few articles in the databases we used during the writing process. The quantity of text is directly proportional to the number of reports available on the anticancer properties of qualified alkaloids.

  • Figures quality is not off the standard and must be improved,

We would like to express our gratitude for your critical view of the figures we used in this review. To meet your expectations, we reduced the amount of background, improved the quality of centrally located graphics and the content of one of the text fields that may have seemed unreadable. We hope that the modifications made will help you appreciate these illustrations with a more approving perspective.

  • Line-16, please italicize In vivo and in vitro and do the said throughout MS,

Changed to italics. However, we leave this issue to the Editors to decide, since according to MDPI requirements they should be written without italics.

  • All scientific names i.e. plants should be in italics and also mention authority names, accordingly.

Thank you very much for your attention, all plant names have been changed to italics.

  • Line-653, Several studies have also been conducted that describe the effect of CHL or its derivatives on gastric cancer cells. Where is several citations?

Thank you for this reasonable remark. We found this sentence as an introduction to the subsection about CHL activity against GC and in the following part of the subsection the mentioned several articles were cited. We understand that it could be unclear, so the sentence was modified and now it's presented as follows: “There are also a few reports of anti-cancer properties of CHL against GC.”

  • 3.2. Chelerythrine wrote in two paragraphs and you just cited three articles? In a similar way, 3.3. Sanguinarine is described in three paragraphs and you had just cited three articles?, very strange.

Thank you for your opinion in this field. In order to make the subsections about CHL and SAN less controversial in reception, we unified their appearance. Currently, each of them is created by one paragraph regardless of the number of articles cited.

  • If possible, please put a table summarizing different biological activities of the said alkaloids in different forms of cancer and/or cancer cell lines.

Thank you for your attention to the topic. In fact, our review lacked a table summarizing the less documented alkaloids, so we introduced two tables according to your suggestions relating to: the type of cancer, the tested cell lines and the documented biological effect. In our opinion, they are an important supplement and summary of the most important reports from the cited articles and together with illustrations form a comprehensive picture of the complex action of alkaloids in cancer therapy.

  • Why only 110 references were these important?

The number of literature items is directly proportional to the number of Articles on the subject of their anticancer properties in the databases we used. After the Reviewer's remark about the poorly cited Introduction, the number of citations increased to 125.

 I hope that the corrections we have made are satisfactory and improve the quality of the Manuscript.

In addition, we would like to inform the Reviewer that the Manuscript, as revised, has been carefully reviewed for the English language used by a senior Professor in the field. He has made numerous corrections in many places. I hope that the current quality of the English language is satisfactory to the Reviewer.

Kind regards,

Anna Duda-Madej

Round 2

Reviewer 2 Report

Comments and Suggestions for Authors

Dear Editors/authors, 

Many thanks for considering my suggestions and the improvements made. 

However, I still have some minor points that must be improved to meet the quality and standard of the journal/MS. 

For example:

1. Introduction first paragraph, please re-write these sentences and cite them accordingly. The human organism in a state of health is characterized by homeostasis, a balance of cells between death and their divisions. Disruptions of this state lead to errors in the recording and reading of cellular DNA, resulting in one population of cells growing abnormally and uncontrollably.

2. Line-32 1st and 3rd must be 1st and so on, please do consider throughout MS,

3. Like in the previous round I suggest you write the author names for example, Coptis chinensis should be Coptis chinensis Franch., please do this throughout MS,

4. For sure, the figures are fine now, however, please indicate in the description of Fig. 2-5 about and lower ↓.

Kind regards, 

Comments on the Quality of English Language

Dear Editor/authors, 

Again, I am not happy with the English language used, please consider English editing, before the onward step. 

Thanks

Author Response

Dear Reviewer,

Thank you very much for reviewing again in such a short time. We have complied with the reviewer's comments that follow:

  1. Introduction first paragraph, please re-write these sentences and cite them accordingly. The human organism in a state of health is characterized by homeostasis, a balance of cells between death and their divisions. Disruptions of this state lead to errors in the recording and reading of cellular DNA, resulting in one population of cells growing abnormally and uncontrollably.

Thank you for pointing us to this place. We did not previously feel that a citation here was necessary, but in accordance with the reviewer's comment, the literature position [1] has been added:

Cooper, G. The Cell: A Molecular Approach. The Development and Causes of Cancer.; 2nd editio.; Sinauer Associates: Sunderland (MA), 2000;

  1. Line-32 1st and 3rd must be 1stand so on, please do consider throughout MS,

We thank the reviewer for his attention. We used ordinal numbers only in the introduction. We have changed their notation accordingly to: 1st, 3rd and 5th.

  1. Like in the previous round I suggest you write the author names for example, Coptis chinensis should be Coptis chinensis, please do this throughout MS,

Thank you very much for this valuable comment. We thought all the names were already spelled correctly after our earlier corrections. However, after deeper investigation, we found that some names are incomplete. We have made the necessary changes.

  1. For sure, the figures are fine now, however, please indicate in the description of Fig. 2-5 about ↑ and lower ↓.

We have followed the reviewer's comment and added an explanation of the meaning of the arrows under each Figure 2-5 and in the Materials and Methods section: ↓ - down-regulation; ↑ - up-regulation.

We sincerely believe that our corrections have improved the quality of our Manuscript, which will meet with the Reviewer's approval.

Kind regards

Anna Duda-Madej
